# Self-care interventions for preconception, antenatal, intrapartum and postpartum care: a scoping review

Phi-Yen Nguyen ![ORCID] ,[1,2] Cassandra Caddy,[2] Alyce N Wilson,[2] Kara Blackburn,[2] Matthew J Page ![ORCID] ,[3] A Metin Gülmezoglu ![ORCID] ,[4] Manjulaa Narasimhan,[5] Mercedes Bonet,[5] Özge Tunçalp,[5] Joshua P Vogel ![ORCID] [2]

[1]Methods in Evidence Synthesis Unit, Monash University, Melbourne, Victoria, Australia
[2]International Development, Burnet Institute, Melbourne, Victoria, Australia
[3]School of Public Health and Preventive Medicine, Monash University, Clayton, Victoria, Australia
[4]Concept Foundation, Geneva, Switzerland
[5]Department of Reproductive Health and Research, World Health Organization, Geneva, Switzerland

**Correspondence to**
Dr Joshua P Vogel;
Joshua.vogel@burnet.edu.au

## ABSTRACT

**Objective** To identify current and emerging self-care interventions to improve maternity healthcare.

**Design** Scoping review.

**Data sources** MEDLINE, Embase, EmCare, PsycINFO, Cochrane CENTRAL/CDSR, CINAHL Plus (last searched on 17 October 2021).

**Eligibility criteria** Evidence syntheses, interventional or observational studies describing any tool, resource or strategy to facilitate self-care in women preparing to get pregnant, currently pregnant, giving birth or post partum.

**Data extraction/synthesis** Screening and data collection were conducted independently by two reviewers. Self-care interventions were identified based on predefined criteria and inductively organised into 11 categories. Characteristics of study design, interventions, participants and outcomes were recorded.

**Results** We identified eligible 580 studies. Many included studies evaluated interventions in high-income countries (45%) and during antenatal care (76%). Self-care categories featuring highest numbers of studies were diet and nutrition (26% of all studies), physical activity (24%), psychosocial strategies (18%) and other lifestyle adjustments (17%). Few studies featured self-care interventions for sexual health and postpartum family planning (2%), self-management of medication (3%) and self-testing/sampling (3%). Several venues to introduce self-care were described: health facilities (44%), community venues (14%), digital platforms (18%), partner/peer support (7%) or over-the-counter products (13%). Involvement of health and community workers were described in 38% and 8% of studies, who supported self-care interventions by providing therapeutics for home use, training or counselling. The most common categories of outcomes evaluated were neonatal outcomes (eg, birth weight) (31%), maternal mental health (26%) and labour outcomes (eg, duration of labour) (22%).

**Conclusion** Self-care interventions in maternal care are diverse in their applications, implementation characteristics and intended outcomes. Many self-care interventions were implemented with support from the health system at initial stages of use and uptake. Some promising self-care interventions require further primary research, though several are matured and up-to-date evidence syntheses are needed. Research on self-care in the preconception period is lacking.

## STRENGTHS AND LIMITATIONS OF THIS STUDY

⇒ Mapping all potential self-care practices across the spectrum of pregnancy care, from preconception, antenatal, intrapartum to postnatal care.

⇒ Use of the new WHO self-care classification system to organise and categorise findings.

⇒ Not screening for primary studies reported within an included evidence synthesis.

⇒ Not including interventions that took place after 42 days post partum, or those that related primarily to care of newborns and breast feeding (which should require dedicated reviews).

⇒ Search limited to published reports and reports written in English only.

## BACKGROUND

Self-care is defined by the WHO as 'the ability of individuals, families and communities to promote health, prevent disease, maintain health, and to cope with illness and disability with or without the support of a health worker'.[1] Self-care may include, but is not limited to, health literacy, self-awareness of physical and mental health, physical activity, healthy eating, good hygiene and rational use of medicines and diagnostics.[2] By extension, self-care interventions are those that can support or promote people's self-care.[3] Self-care interventions can act to complement or even broaden delivery of health services, ideally improving health outcomes and care experiences.[4 5] Humanitarian crises, emergencies and pandemics introduce additional challenges to the provision of routine healthcare services.[6 7] In this context, self-care interventions can help ensure health and well-being is maintained with reduced in-person contact.[8]

In 2021, WHO published a guideline on self-care interventions for health and well-being, which included 13 recommendations on selected self-care interventions to improve preconception, antenatal, intrapartum and

postpartum care.[3] By encouraging women to engage in these self-care interventions, the overall efficiency of healthcare delivery can be enhanced. In addition, there are interventions for pregnant and postpartum women that are not currently included in WHO's self-care guideline, but have the potential of transitioning into a self-care paradigm. Examples include prescription medicines that can be made available over-the-counter (OTC), self-collection of samples for testing or home testing kits for certain diseases. Some systematic reviews have been conducted on self-care interventions in pregnancy[9–11]; however these previous reviews focused on a few selected interventions and outcomes, and mainly the antenatal or postpartum periods. No previous review has explored the broad and heterogeneous body of research across preconception, antenatal, intrapartum and postpartum periods, in order to identify those that may be effective or applicable from a self-care perspective. We therefore aimed to identify all current and emerging self-care interventions via a scoping review that have been evaluated in the context of preconception, antenatal, intrapartum and postpartum care.

## METHODS

The scoping review was conducted in accordance with the methodological guide for scoping reviews by Peters and colleagues,[12] and reported in accordance with the Preferred Reporting Items for Systematic Reviews and Meta-Analyses guideline for Scoping Reviews[13] (online supplemental file 1). The protocol was registered online at https://doi.org/10.17605/OSF.IO/97X25 (online supplemental file 2).

### Electronic searches

The search strategy was developed in consultation with an information specialist (online supplemental file 3) and combined self-care concepts with terms related to preconception, antenatal, intrapartum and postpartum care, based on the framework proposed by Narasimhan and colleagues.[4] The terms and concepts used were informed by previous reviews of pregnancy care and self-care for other health conditions[14–16] and by inputs from team members with clinical and public health background. The following databases were systematically searched to identify published studies from inception to 17 October 2021: MEDLINE, Embase, EmCare and PsycINFO (via Ovid), CENTRAL/Cochrane Database of Systematic Reviews (via Cochrane Library) and CINAHL Plus (via EBSCOhost). The last date of search was 17 October 2021. Reference lists of included studies were manually screened to identify other relevant studies. Grey literature was not searched after considering the volume and variable quality of such evidence.[17]

### Study selection

Two reviewers (P-YN and CC) first independently screened all titles and abstracts to exclude irrelevant studies, and then independently reviewed potentially eligible full texts based on predetermined eligibility criteria. All stages of screening were conducted using Covidence software.[18] Any disagreement was resolved by discussion or adjudication by a third reviewer (JPV).

The eligibility criteria for inclusion were: (A) Participants: women preparing to get pregnant, currently pregnant, giving birth or in the postpartum period (up to 42 days after birth), of any age. Studies in which partners or family members carried out self-care for the pregnant woman, were also eligible. Interventions for abortion were not eligible. We also excluded interventions that were only directed at newborns and postnatal breastfeeding interventions for improving neonatal outcomes, though antenatal programmes related to preparation for the postnatal period were eligible. (B) Interventions: for this review, a self-care intervention was defined as any tool, resource or strategy designed to promote or facilitate self-care by a woman intending to get pregnant, or a currently pregnant, labouring or in the postpartum period, for the purpose of improving the quality or coverage of maternity healthcare, and/or improving her health, well-being and care experiences. To be eligible, interventions must have been aimed at reducing risks of pregnancy, childbirth or postpartum complications and/or enabling a positive pregnancy, childbirth or postpartum experience. Each intervention was evaluated independently by two reviewers for inclusion based on a set of prespecified principles (online supplemental file 4). (C) Comparator: placebo, no treatment or standard care; single-arm interventional studies were also eligible. (D) Outcomes: outcomes related to maternal physical, psychological or emotional health, behavioural change affecting a mother's own health or diagnostic accuracy of tests for women. Studies reporting economic outcomes only were not included, and neither were feasibility studies evaluating only retention rates and acceptability. (E) Types of studies: randomised trials, quasi-experimental studies and cohort studies, scoping reviews, systematic reviews, meta-analyses and overviews of systematic reviews. Systematic reviews and overviews must have stated some form of search strategy and presented all included studies in a systematic manner to be eligible for inclusion. Ongoing studies (defined as trials with protocols published from 2019 onwards which were not completed) were eligible; no restriction on date of publication was applied otherwise. Case series, case studies, case reports, dissertations and theses, reports published as abstracts only, correspondence letters, editorials and qualitative studies were excluded. Only studies with full texts written in English were eligible.

### Data collection and analysis

Two reviewers independently extracted data using a standardised extraction form we developed to suit the review (online supplemental file 5). Data extracted included characteristics of each study's population, interventions and outcomes measured. Study designs were classified as

evidence syntheses, interventional studies, observational studies or diagnostic accuracy studies. Countries where studies took place were classified based on the World Bank 2021 country income levels.[19] For population characteristics, we recorded whether the studies included women under the age of 18 and whether those with high-risk pregnancies were included. For each study, the self-care activities that the intervention aimed to promote or facilitate, and the period of pregnancy during which the intervention took place (preconception, antenatal, intrapartum or postpartum period) were recorded. An inductive thematic analysis process was undertaken.[20] After all self-care interventions were identified, we analysed their descriptions and identified themes among groups of interventions. For example, interventions involving taking dietary supplements or adjusting diet intakes were grouped under the theme 'Diet and nutrition'. These themes were refined until all interventions were accounted for, creating the final set of self-care categories. In addition, we matched the interventions to the categories listed in WHO's classification of self-care interventions for self-carers and caregivers.[21] We also recorded the venues of access to self-care, that is, the setting or location where participants were orientated to or carried out self-care, based on this classification framework.[21] Lastly, we recorded whether any external support person was required to promote or facilitate self-care. These information were recorded for primary studies only, as evidence synthesis studies may include studies with different characteristics. Any disagreement was resolved via consensus or adjudication by a third reviewer (JPV). In line with scoping review methodology,[12] quality assessment of included studies was not performed. Descriptive statistics (frequency and percentage) were used to report study, participant and intervention characteristics. Studies that feature more than three self-care interventions were analysed and reported separately. R V.4.1.2 was used to calculate the frequencies of various intervention characteristics, participant characteristics and outcomes for each self-care intervention, aggregate these frequencies by self-care categories and present them as bar charts.

### Patient and public involvement statement

Patients were not included in the design, conduct and plans for dissemination of results of this review.

### RESULTS

A total of 12 215 records from six databases were retrieved, together with an additional 182 records from manual reference screening (figure 1). In total, 10 712 reports were excluded at the title/abstract screening stage and another 743 reports at full-text screening (online supplemental file 6). The final sample consisted of 580 unique studies (616 reports) (online supplemental file 7), 18 (3%) of which were ongoing studies.

### Characteristics of included studies

Among 580 studies, two-thirds were interventional studies (n=365, 63%), many of which reported results

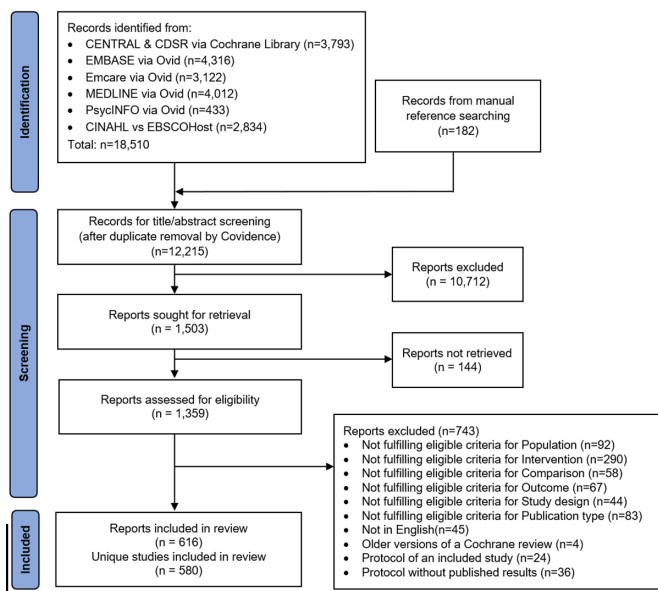

**Figure 1** Preferred Reporting Items for Systematic Reviews and Meta-Analyses flowchart of record retrieval and selection.

of randomised trials (n=306, 53%). Evidence syntheses (n=165) accounted for 28% of all studies, out of which 109 were meta-analyses (19%). Observational studies were less common (n=32, 6%) and were mainly retrospective or prospective cohort studies (n=25, 4%). Study designs used were largely similar across all self-care categories, with some exceptions (figure 2A, online supplemental file 8). In self-management of medication, the proportion of non-randomised studies (n=7, 41%) was similar to that of randomised trials (n=6, 35%). In self-testing and self-collection of samples, many studies (n=10, 56%) were diagnostic accuracy studies.

Most studies took place in high-income countries (n=260, 45%), followed by middle-income (n=146, 25%) and low-income countries (n=11, 2%); 163 studies (28%) were reviews with no geographical restriction, and usually included studies from multiple countries. Countries with the most studies were the USA (n=86, 15%), Iran (n=46, 8%), UK (n=33, 6%) and Australia (n=30, 5%). Self-care interventions most commonly took place in the antenatal (n=439, 76%) and postpartum periods (n=119, 21%). Few studies related to the preconception (n=21, 4%) or intrapartum periods (n=62, 11%) (figure 2B, online supplemental file 8).

Thirty-eight studies (7%) included women under the age of 18. One-third of included studies (n=170, 29%) included women with high-risk pregnancies. High-risk pregnancies were variably defined—the most common definitions included pregnant women with gestational or existing diabetes (n=54, 9%), obesity or overweight (n=35, 6%) and ex-smokers/current smokers (n=31, 5%).

### Self-care interventions and their characteristics

Among included studies, 35 studies (6%) described a complex intervention involving more than three different

## A Distribution by study design

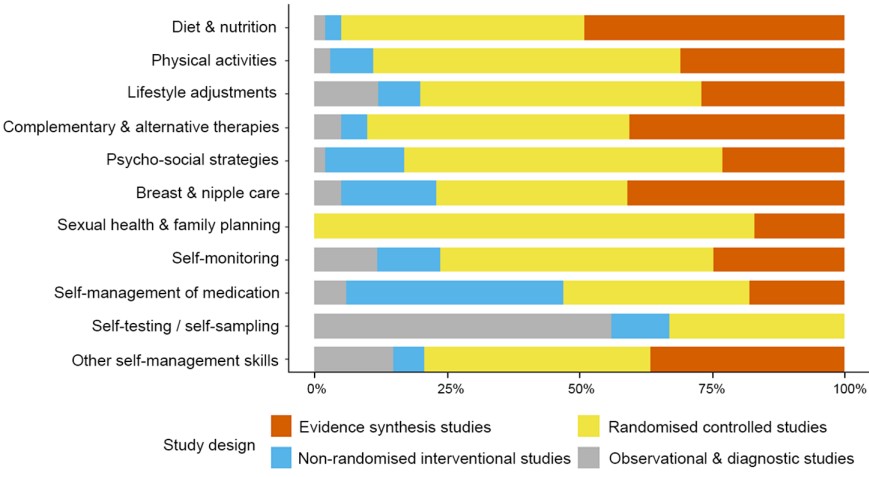

## B Distribution by time of intervention

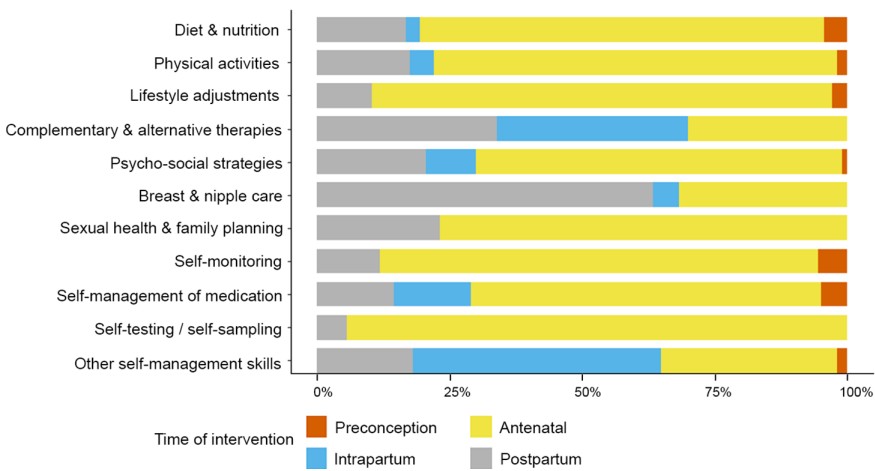

**Figure 2** Characteristics of included studies.

self-care activities (online supplemental file 9). The remaining 545 studies (94%), which featured up to three self-care activities, constituted the final sample for identification of self-care interventions. A total of 112 unique self-care interventions were identified and organised into 11 broad categories (figure 3; online supplemental file 10). The most common self-care categories identified included diet and nutrition (n=143/545, 26%), physical activity (n=132, 24%), psychosocial strategies (n=97, 18%) and lifestyle adjustments (n=90, 17%). Few studies featured self-care interventions under the categories of sexual health and family planning (2%), self-management of medication (3%) and self-testing/sampling (3%). All identified interventions demonstrate various degrees of interaction with the health system,[4] from little to no direct contact (self-awareness/regulation), some interaction (self-management of health conditions) and close linkage with the health system for follow-up (self-testing and diagnosis) (figure 3).

The identified interventions correspond to a range of categories from WHO's classification of self-care interventions (online supplemental file 11). The most common WHO categories were interventions for improving capacity and promoting autonomy in self-care (n=325, 56%), self-care prevention including risk avoidance and support for physical and mental health and wellbeing (n=296, 51%) and self-care for short-term health conditions (n=189, 33%). Some WHO categories did not feature among included studies, such as on-demand information services for health information, diagnostic devices in community locations, online symptom checkers for diagnosis and interventions enabling individuals to identify location of health facilities/structures. Notably, we identified no studies that targeted self-care outside the context of a specific health issue, such as campaigns to promote awareness about self-care or programmes to improve health and digital literacy.

Among 545 studies under analysis, self-care was accessible through several avenues as described in the WHO classification framework.[20] These avenues include health facilities (n=238/545, 44%), digital platforms (n=97, 18%), community and non-facility venues (n=74, 14%), partner or peer support (n=37, 7%) or OTC products, for example, nutritional supplements, topical products or

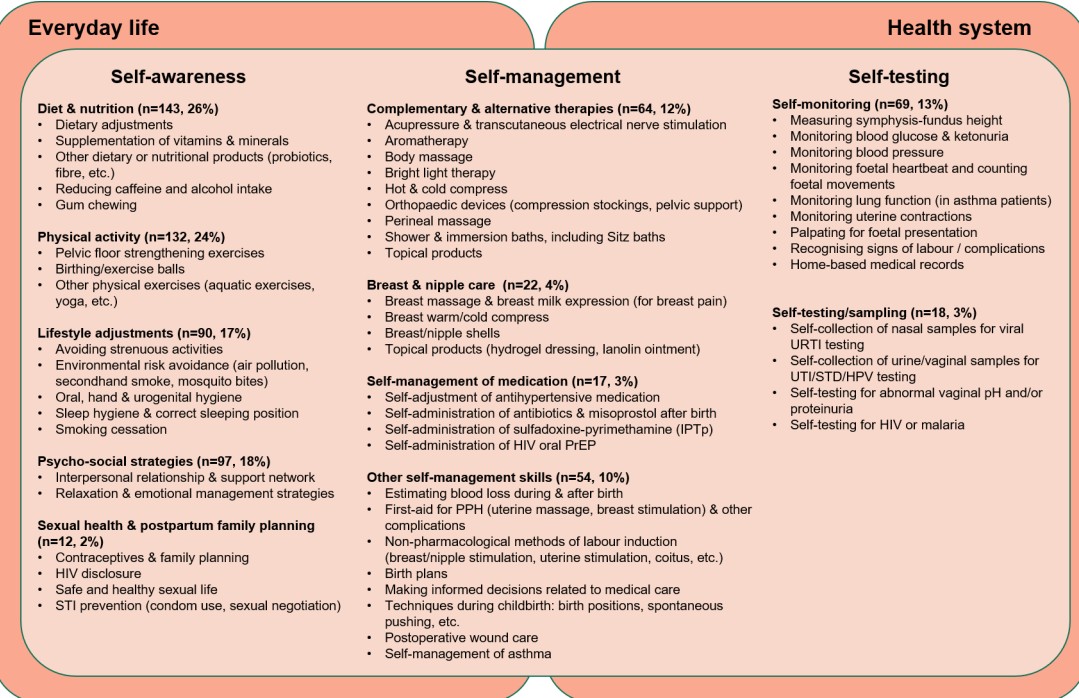

*Some studies feature more than one self-care activities, hence the percentages do not total to 100%.*
*Some primary studies (n=45, 8%) are also included in evidence synthesis studies featured in this review.*
*For further details on each intervention, see Supplementary File S10.*

**Figure 3** All self-care interventions identified from included studies and their interactions with the health system. HPV, human papillomavirus; IPTp, intermittent preventive treatment of pregnancy; PPH, postpartum haemorrhage; STD/STI, sexually transmitted diseases/infections; PrEP, pre-exposure prophylaxis; URTI, upper respiratory tract infection; UTI, urinary tract infection

support devices (n=69, 13%). In 80 studies (15%), there was insufficient detail to determine that the aforementioned modes of access were used. No studies were identified where the intervention took place at the workplace, in schools or in emergency or humanitarian settings.

Across the self-care categories, provision of information and demonstration of self-care were most commonly done in health facilities, especially for self-testing (94% of studies in this category) and self-monitoring (58%). Provision of similar services outside health facilities, such as via community health workers or at community-based venues, was mainly for the support of self-management of medication (41%) and physical activity (28%). Home-based self-care (ie, direct access to self-care resources and information at home) was less frequent, mainly in the categories of breast or nipple care (23%) and lifestyle adjustments (20%). Digital platforms were popular in some categories, notably self-monitoring (51%). Use of OTC products was common in complementary/alternative therapies (39%) and breast and nipple care (27%). Lastly, peer or partner support was relatively less common across categories (7%), and we observed that partner support (1%) was relatively less frequent than support from peers (eg, other women, members of support groups) (6%) (figure 4A, online supplemental file 8).

A large number of studies described the involvement of health workers in promoting or facilitating self-care (n=208/545, 38%), especially for sexual health and family planning (67% of studies in this category), psychosocial strategies (45%) or physical activity (44%) (figure 4B, online supplemental file 8). Few studies mentioned involvement of trained community health workers (n=41, 8%), non-trained community members (n=25, 5%) or family members (n=3, 1%).

### Types of outcomes in self-care studies
Included studies used a diverse range of outcomes, including both maternal and neonatal outcomes (online supplemental table 1). The most commonly measured ones were neonatal outcomes (birth weight, Apgar scores, preterm birth, admission to neonatal intensive care unit and other neonatal complications) (n=177, 31%); maternal mental well-being and quality of life (n=150, 26%), and labour outcomes (duration of labour stages, labour pain, mode of delivery or labour induction methods) (n=125, 22%). A total of 147 studies (25%) measured behavioural outcomes, mainly related to reductions in risky behaviours such as smoking or alcohol consumption. Maternal mortality (n=20, 3%) and perinatal mortality (n=67, 12%) were less commonly evaluated.

## A Distribution by venue of access to intervention

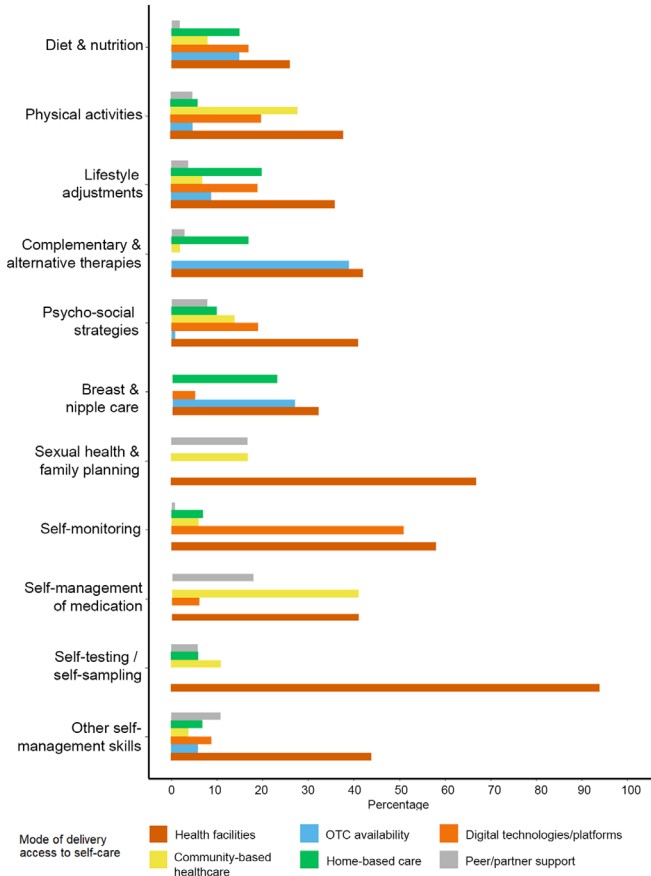

## B Distribution by persons providing external support

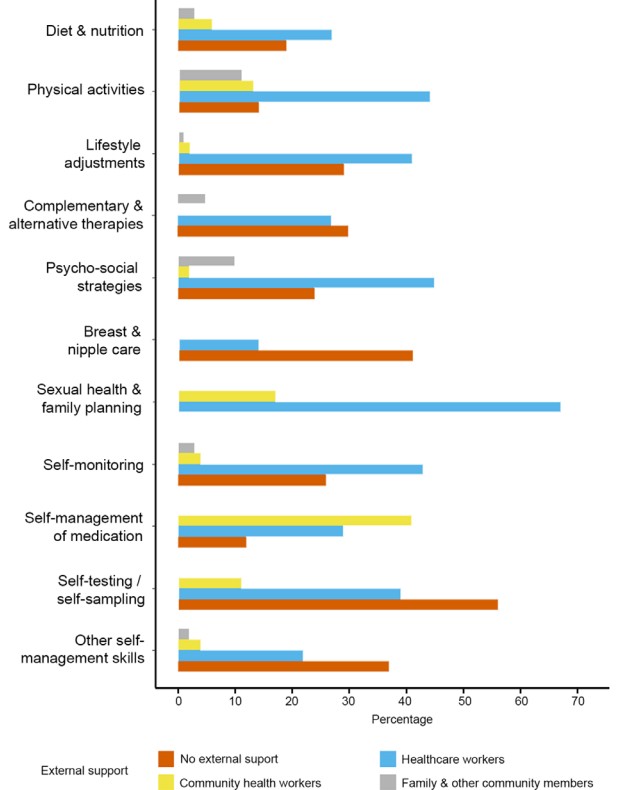

**Figure 4** Characteristics of self-care interventions. OTC, over-the-counter.

## DISCUSSION
### Main findings

In this scoping review, 580 studies were identified pertaining to 112 self-care interventions across preconception, antenatal, intrapartum and postpartum periods. These interventions encompassed multiple modalities, settings and levels of provider involvement. Most self-care interventions required close engagement with health facilities, in which healthcare workers or community health workers would teach self-care skills or provide therapeutics for home use. Previous studies also suggest that healthcare services are a critical starting point for successful self-care approaches.[22–24] As such, self-care should be viewed as an extension of, and not a separate entity to, formal healthcare services. Our findings also reflect a broader trend towards task-sharing in order to increase maternity care coverage.[7 25] While the majority of self-care interventions were implemented in the antenatal period, many of these studies aimed to equip pregnant individuals with self-care skill via workshops or counselling, which can be used in the intrapartum (eg, relaxation methods during labour) or postpartum periods (eg, pelvic floor exercises, breast and nipple care). Few studies examined the potential benefits of self-care at the preconception period. Few interventions featured digital platforms (eg, websites, mobile applications); those identified were mostly used for communication between the pregnant individual and a remote health worker, or as a platform for the pregnant individual to record her medical information and receive automatic, individualised health advice. While digital technologies seem to currently play a limited role, they have great potential to catalyse and promote maternity-related self-care.[26]

Most studies concentrated on self-care interventions related to lifestyle modification—such as diet, nutrition/supplement intake, physical activity and sleep habits—as well as strategies to optimise mental and social well-being. The established evidence base for lifestyle interventions was also reflected in previous systematic reviews on self-care for pre-eclampsia and gestational diabetes.[9 11] This might be attributable to a perceived lower risk of harm when conducting trials investigating lifestyle interventions, compared with clinical or pharmacological interventions. Moreover, lifestyle interventions tend to take place in their own living environments and in their own time, and hence may be more amenable to self-care approaches. Conversely, there were relatively few studies that explored self-monitoring, self-testing or self-management of medications or pregnancy complications. While studies of these self-care approaches can be complex or technically challenging, they could provide solutions to improving care in remote and low-resource settings.[6] While advanced distribution and self-administration of misoprostol for postpartum haemorrhage prevention has been well-explored,[27–29] there was limited or no research on similar approaches to other therapeutics, such as use of anthelminthics, malaria prophylaxis, antimalarials, antiretroviral therapy or vitamin and micronutrient supplementation. Although many studies reported on the prevalence and determinants of self-medication during pregnancy,[30–32] none investigated interventions aimed at promoting safe self-medication practices or how to seek information when self-medicating. There were also few self-care interventions aimed at communicable disease detection and prevention during or after pregnancy. Future research is warranted to investigate how these aspects of clinical care during and after pregnancy could be approached from a self-care perspective.

We identified studies describing self-testing options for urinary tract infections and HIV, but few or no studies on malaria, syphilis or chlamydia self-testing, for which rapid test technologies are available but not yet scaled up or tested in pregnancy.[33–35] Similarly, self-collection of vaginal samples for reproductive tract infections is an emerging alternative to clinic-based sampling.[36] While some studies have investigated diagnostic accuracy of self-sampling for reproductive tract infections in pregnancy, evidence on their benefits, harms and effective linkages with health systems is currently lacking.[37] Lastly, we identified self-care studies on insecticide-treated nets for malaria, mostly from the sub-Saharan Africa region. However there was little research on other low-cost strategies amenable to a self-care approach (indoor residual spraying and insect repellents), or from other malaria-endemic regions such as Asia and Latin America.[38]

Last but not least, we found no eligible studies on traditional and Indigenous self-care practices. While important qualitative research has been conducted on this topic,[39 40] we hypothesise that there is a relative lack of studies exploring the health effects of these types of self-care practices. Future research on these topics could also provide new opportunities for improving cultural safety of maternity care services.

### Strengths and limitations

While there are systematic reviews for some self-care practices, this is the first scoping review that maps all potential self-care practices across the spectrum of pregnancy care, from preconception, antenatal, intrapartum to postnatal care. This scoping review provides a comprehensive summary of the state of research on this topic, and lays the foundation for future systematic reviews and primary research. We analysed studies based on multiple characteristics, as well as applying the new WHO self-care classification system to organise, categorise and facilitate interpretation of findings.

This scoping review was intended to present an overview of the research landscape, and not to critically appraise or meta-analyse available evidence. The total number of studies identified for a given self-care intervention may not correlate with evidence quality, or its effectiveness. We reported the results for complex interventions (ie, interventions with more than three different self-care practices) separately (online supplemental file 9), since segregating each complex intervention into multiple individual interventions would overestimate

their representation in the sample and disregard potential synergy between the self-care components. Besides, because these studies did not introduce any new self-care intervention, or a substantive number of studies for any single self-care intervention that we had already identified, it is unlikely that including them into the main analysis would change our results and conclusion. Although the number of such studies is small (n=35, 6%), they represent how self-care can be implemented in real-life, and future reviews should study their effects using appropriate methods, for example, component network meta-analysis.[41 42]

Some limitations must nonetheless be acknowledged. We did not formally screen those primary studies reported within an included evidence synthesis, and we assessed intervention and outcomes characteristics based on the review's summary description of the included studies. It is possible that the number of primary studies was there underestimated, or the characteristics of individual primary studies might differ. Our assessment found 45 primary studies (8% of all included reports) that were also included within an evidence synthesis included in this review. Given their small number, we consider it unlikely that these overlapping studies would substantively affect the overall results. Hence, the distribution of study designs (figure 2A, online supplemental file 8) should be considered in addition to the volume of evidence available for each intervention. We opted to focus on maternity-related self-care, thus excluding interventions that took place after 42 days post partum, or those that related primarily to care of newborns and breast feeding. Our research objectives also mean that we did not include qualitative studies. These exclusions do not discount their importance, but rather acknowledge that these topics require separate, dedicated reviews. Our search of published reports means that self-care interventions that have only been evaluated in unpublished studies or grey literature would not have been identified. With 45 reports excluded as not written in English (3.3% of all full-text reports), language bias is possible.

### Future research
This review was conducted for the purpose of guiding further evidence synthesis activities and assisting the WHO with guideline development in this research area. This scoping review identified two distinct groups of self-care interventions, which have different pathways forward. The first group are those interventions with a substantial number of primary studies, where no systematic review has been conducted or only outdated systematic reviews exist. For these interventions—such as use of clean cookstoves to reduce adverse birth outcomes associated with indoor air pollution, self-care interventions to reduce postpartum depression, or self-collection of vaginal samples to test for sexually transmitted diseases—up-to-date systematic reviews are warranted to inform guidelines, clinical practice and self-care programmes. The second group are interventions with limited or no

primary studies, or where available primary studies were not designed to fully deliver the intervention in a self-care approach. For these interventions, well-designed primary studies that compare a self-care approach with standard care are needed. Examples include self-testing for proteinuria, or self-measurement of vaginal pH to detect early signs of bacterial vaginosis. Some self-care interventions may be particularly useful in limited-resource settings, and thus should be evaluated in such settings.

### CONCLUSION
Self-care interventions during preconception, antepartum, intrapartum and postpartum periods are diverse in scope and implementation characteristics. Many self-care interventions were implemented with support from the health system at initial stages of use and uptake. Some promising self-care interventions require further primary research, though several are more matured and up-to-date evidence syntheses are needed. Research on self-care in the preconception period is lacking. Future research should explore task-sharing from clinical care to home or community settings, or effective self-management of complications where access to care is unavailable.

**Acknowledgements** We thank Steve McDonald (Cochrane Australia) for his guidance in formulating the search strategy.

**Contributors** The corresponding author attests that all listed authors meet authorship criteria and that no others meeting the criteria have been omitted. P-YN and JPV conceptualised the study, wrote the protocol and statistical plan and organised collaborative meetings. P-YN and CC screened the literature and extracted the data. P-YN analysed the data, wrote the original draft and revised the paper. MJP advised on the design of the study. CC, AW, KB, MJP, AMG, MN, MB, OT and JPV participated in technical and clinical discussions, and reviewed and revised the paper. JPV is the guarantor of the study.

**Funding** This project was funded by UNDP/UNFPA/UNICEF/WHO/World Bank Special Programme of Research, Development and Research Training in Human Reproduction (HRP), Department of Sexual and Reproductive Health and Research, WHO (# PO 202724071). This work was also supported, in part, by the Bill & Melinda Gates Foundation INV-001304. While members of our authorship team (MN, MB and OT) are employed by HRP/WHO, the funder organisations had no role in the design and conduct of the study; collection, management, analysis and interpretation of the data; preparation, review or approval of the manuscript; and decision to submit the manuscript for publication.

**Disclaimer** The author is a staff member of the World Health Organization. The author alone is responsible for the views expressed in this publication and they do not necessarily represent the views, decisions or policies of the World Health Organization.

**Competing interests** The authors declared no competing interests.

**Patient and public involvement** Patients and/or the public were not involved in the design, or conduct, or reporting, or dissemination plans of this research.

**Patient consent for publication** Not applicable.

**Ethics approval** Not applicable.

**Provenance and peer review** Not commissioned; externally peer reviewed.

**Data availability statement** Data are available in a public, open access repository. Data supporting the findings and conclusion of this study are included in this published article and its supplementary files. The underlying data set and analytical code are available at osf.io/26f7y.

**ORCID iDs**
Phi-Yen Nguyen http://orcid.org/0000-0002-0476-3385
Matthew J Page http://orcid.org/0000-0002-4242-7526
A Metin Gülmezoglu http://orcid.org/0000-0003-4674-0998
Joshua P Vogel http://orcid.org/0000-0002-3214-7096

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
