## [Reviewer comments · BMJ Open]

ARTICLE DETAILS

TITLE (PROVISIONAL)	Self-care interventions for preconception, antenatal, intrapartum and postpartum care: a scoping review
AUTHORS	Nguyen, Phi-Yen; Caddy, Cassandra; Wilson, Alyce; Blackburn, Kara; Page, Matthew; Gülmezoglu, Ahmet Metin; Narasimhan, Manjulaa; Bonet, Mercedes; Tunçalp, Özge; Vogel, Joshua

VERSION 1 – REVIEW

REVIEWER	Kadar , Kusrini S. Universitas Hasanuddin, Faculty of Nursing
REVIEW RETURNED	18-Nov-2022

GENERAL COMMENTS	This is a massive data of paper to explore self-care intervention for preconception, antenatal, intrapartum, and postpartum care. Some suggestions in the presentation of the data, please consistently present the data with percentages or total numbers. Another suggestion is in the paper it is not clear what kind of self-care intervention is for each stage of pregnancy. There is no mapping to see interventions for each stage. Thus, it is much better if the authors can present this in one table or figure. Figure 2 presents only the self-care interventions identified from included studies and their interactions with the health system but does not clearly show the intervention per stage of pregnancy.
--

REVIEWER	Sarmiento, Ivan McGill University, Family Medicine
REVIEW RETURNED	29-Nov-2022

GENERAL COMMENTS	Thank you for the opportunity to read and comment on this manuscript, which reflects an extraordinary amount of work and effort. The study provides a comprehensive description of self-care interventions for preconception, pregnancy, childbirth and postpartum available in the published literature. The manuscript requires minor revisions to explain its strengths and limitations and to present more details on the classification procedure to create the categories of interventions. Because the review included a literature synthesis, the authors need to establish and discuss possible duplication of references and the impact of using aggregated data. Authors need to explain how they handled the literature reviews. The results depend on the proportion of studies in each category. Therefore, results could change significantly if there are duplications or a review includes multiple studies. I encourage the authors to mention another set of self-care practices derived from indigenous traditional knowledge in the discussion.[1–3] These practices are often ignored in the literature
--

	and even more so by international organisations that have not adopted culturally safe approaches to health. These practices are not part of complementary or alternative therapies (this category is already in the review) but are part of the repertoire of traditional birthing systems.[4] Ignoring the existence of traditional practices perpetuates a biased view of self-care practices as interventions to promote task-shifting and sharing from clinical settings to homes but neglects the potential of learning from other ways of knowing. Abstract Why are 45% of studies the majority? Indicate the included studies at the beginning of the results section. Strengths The first two strengths are characteristics of a scoping review. They are not the strengths of this study. The abstract says the authors followed an inductive approach to categorisation, but in this section, they say they used categories from the WHO classification. Please correct this discrepancy. Limitations Exclusion criteria are not necessarily limitations of the study (P5, line 81). Please think of characteristics of the studies that will limit their conclusions within the scope of the review. Methods Please indicate the registry's date and the online platform's name and provide the link in brackets. Page 5, line 99. Please, consider reorganising the paragraph to connect the inclusion criteria to the first sentence. Page 6, line 103. Please briefly clarify this criterion "Studies involving partners or family members in maternal care were also eligible." Page 7, lines 129 to 131. Please revise for clarity. Page 7, lines 137 to 142. Please, briefly elaborate to explain how you conducted the inductive thematic analysis. It is unclear why you used another set of categories for interventional and observational studies. Page 7, line 142. Please consider moving this sentence to the second place in the paragraph. Page 7, line 144. Please, briefly explain why you excluded complex interventions from the final analysis. Is this a limitation to mention after the abstract and in the discussion? Page 7, lines 145 to 147. However, you still need to explain what the final analysis was. What analyses did you do in R? Please, clarify Please, explain how you handled the literature reviews to avoid duplication of results with other included studies or missing aspects of the studies that those reviews did not consider. This might be a significant limitation because you used the proportion of studies to report your results. Methods Figure 2 was almost impossible to read. You may need to reorganise it. One option is to avoid duplicating the label of categories in module B). Also, you must indicate the number of studies for each row. Page 8, line 166. Could you explain why the numbers do not add to 580? Is it due to literature reviews or studies not reporting a country? Did you consider the countries of the studies in the literature reviews? Please, be explicit about the denominator. Page 10, line 179. Why not include these studies in a category for complex interventions instead of eliminating them? Please, in the limitations, discuss the impact of this decision.
--	--

	Page 9, lines 200 to 202. Is this potentially a result of the search strategy? Page 10, lines 207 to 210. There are no reasons to make these assumptions. Please only report that in 80 studies (15%), there was insufficient detail to determine that the aforementioned modes of access were used. Figure 4 is almost impossible to read. The authors should consider making this figure a table offered as an additional supplementary file. Page 10, line 220. Please, provide the numbers for partner support in relation to peers. Page 11, lines 232 to 233. Please, clarify that you used all the studies for this section. The note under Table 1 will not be enough to facilitate readers to see what you did. Also, please refrain from repeating the numbers and percentages that you already have in the table. Discussion Page 13, lines 23 to 26. Do the authors consider health system and formal health care the same? Please revise the sentence. Page 13, line 56. What elements do you have to think this likely reflects a more well-established evidence base for lifestyle modification interventions, generally associated with a lower risk of harm? The sentence might be unnecessary. Please, provide a reference or eliminate it. Strength and limitations This section has similar problems to those described after the abstract. Please, take those comments into account in this section. 1 Raman S, Nicholls R, Ritchie J, et al. Eating soup with nails of pig: thematic synthesis of the qualitative literature on cultural practices and beliefs influencing perinatal nutrition in low and middle income countries. BMC Pregnancy Childbirth 2016;16:192. doi:10.1186/s12884-016-0991-z 2 Sarmiento I, Paredes-Solís S, Dion A, et al. Maternal health and Indigenous traditional midwives in southern Mexico: contextualisation of a scoping review. BMJ Open 2021;11:e054542. doi:10.1136/bmjopen-2021-054542 3 Raman S, Nicholls R, Ritchie J, et al. How natural is the supernatural? Synthesis of the qualitative literature from low and middle income countries on cultural practices and traditional beliefs influencing the perinatal period. Midwifery 2016;39:87–97. doi:10.1016/j.midw.2016.05.005 4 Jordan B. Birth in four cultures: A crosscultural investigation of childbirth in Yucatan, Holland, Sweden, and the United States. 4th ed. Long Grove, IL: : Waveland Press 1993.
--	---

REVIEWER	Stegers, Eric Erasmus MC
REVIEW RETURNED	29-Nov-2022

GENERAL COMMENTS	The review was drafted by dr. Lenie van Rossem (l.vanrossem@erasmusmc.nl) and accorded by Prof. Eric A.P. Steegers Comments; This manuscript concerns a scoping review on self-care interventions around (pre and post) pregnancy care. It is a relevant topic given the high burden on health care, where self-care can contribute in the solution for this. In general the paper is well-
--

	written, and the reporting –especially regarding the choices made and the methodology- is very thorough. Table 1, and figures 2 and 3 are well-designed to give a clear overview of the findings. My main feedback to improve the paper is about the clarity of the wording, and the structure/depth of the discussion. I first read the abstract, and then it was not completely clear to me what aspects were being reviewed. This became clear when I read the paper. Please see specific comments below. Abstract:  • Objective&Results: I have some difficulty with the wording of the objective and results. It seems that the objective is to merely identify the number of apps for selfcare, categorized per period. By using the word evidence in the results, it might give the idea that the effectiveness of the app is being studied. • Results: Could there be a discrepancy between the number of included studies in the abstract (545) versus the number of included studies in the main paper (580)? Background:  • ‘Some systematic reviews have been....pregnancy [9-11]’: I would suggest to reword the sentence. It now seems that you are repeating those reviews, but they have actually been performed for specific outcomes. I think it is important to add that as a detail. Methods:  • How did you deal with potential overlap between systematic reviews and original studies? Results:  • Type of outcomes: Can you extend upon the types of outcomes? I was a bit surprised to see some of those outcomes being used for self-care (such as Apgar scores?). • To me, the ‘study design’ aspects of the included studies seem not to be of added value as this review does not review the effectiveness of the apps, or the quality of the studies. Can you explain why you have included this? Discussion:  • Main findings: the main findings start with a summary of the results, but from ‘Primary healthcare services are....’ it seems more about interpretation of the findings. I would suggest to separate these. • Can you extend on the meaning of ‘digital technologies’? • ‘Most studies concentrated on self-care interventions related to lifestyle modification.... This likely reflect a more well-established evidence for lifestyle modification interventions, generally associated with a lower risk of harm’. In addition, I think that lifestyle interventions are eminently suitable for self-care, as lifestyle takes place all day and in the own environment. • Can you extend on your ideas for future research? • I think a comparison with the previous 3 reviews that were referenced in the introduction can be a valuable addition to the Discussion.
--	---

VERSION 1 – AUTHOR RESPONSE

Reviewer: 1

Dr. Kusriani S. Kadar , Universitas Hasanuddin

This is a massive data of paper to explore self-care intervention for preconception, antenatal, intrapartum, and postpartum care.

Thank you for your feedback and comments to improve this paper.

R1Q1. Some suggestions in the presentation of the data, please consistently present the data with percentages or total numbers.

We had adopted a consistent principle in presenting the data. Where possible, we presented both N and % (e.g. L172-179). When there is a change in denominator, we presented the denominator at the start of the paragraph or sentence (e.g. L172, L199), in which case the it is expected that the new denominator will apply to the remaining N and % figures in the paragraph. We have checked the reported data in the tables and narrative and can confirm it is accurate and consistent with this approach.

R1Q2. Another suggestion is in the paper it is not clear what kind of self-care intervention is for each stage of pregnancy. There is no mapping to see interventions for each stage. Thus, it is much better if the authors can present this in one table or figure. Figure 2 presents only the self-care interventions identified from included studies and their interactions with the health system but does not clearly show the intervention per stage of pregnancy.

Figure 2B summarises the distribution of self-care interventions across each stage of pregnancy, for each category of self-care interventions. We added a column in Supplementary File S10 to show the stage of pregnancy where each self-care intervention has been implemented. Unfortunately, due to journal limits, we cannot create another table or figure to present the data.

Reviewer: 2

Dr. Ivan Sarmiento, McGill University, Universidad Del Rosario

Thank you for your feedback and comments to improve this paper.

R2Q1. Because the review included a literature synthesis, the authors need to establish and discuss possible duplication of references and the impact of using aggregated data. Authors need to explain how they handled the literature reviews. The results depend on the proportion of studies in each category. Therefore, results could change significantly if there are duplications or a review includes multiple studies.

Thanks for raising this point. We have revised this section, with regards to two limitations. Firstly, primary studies in systematic reviews were not screened for inclusion and can be under-represented in this scoping review. We added this point under Discussion (“We did not formally screen those primary studies reported within an included evidence synthesis, and we assessed intervention and outcomes characteristics based on the review’s summary description of the included studies. It is possible that the number of primary studies was there underestimated, or the characteristics of individual primary studies might differ.”; L329-333).

Secondly, there was some overlapping, in terms of an individual study being included for a given intervention, and a systematic review for that intervention also including that the individual study. We have determined that the number of overlapping primary studies was 45 studies (8%), and stated this under the footnote of Figure 3 and Supplementary File S10 (“Some primary studies (n=45, 8%) are also included in evidence synthesis studies featured in this review.”). We also added a note on this in the Discussion: “Our assessment found 45 primary studies (8% of all included reports) that were also included within an evidence synthesis included in this review. Given their small number, we consider it unlikely that these overlapping studies would substantively affect the overall results. Hence, the distribution of study designs (Figure 2A, Supplementary File S8) should be considered in addition to the volume of evidence available for each intervention” (L333-337). In addition, to assist readers with interpreting the overall results, in Supplementary File S10, we provide the number of primary studies and number of evidence synthesis studies for each intervention. We believe that this approach is more sensible (and more inclusive) than excluding overlapping studies.

R2Q2. I encourage the authors to mention another set of self-care practices derived from indigenous traditional knowledge in the discussion.[1–3] These practices are often ignored in the literature and even more so by international organisations that have not adopted culturally safe approaches to health. These practices are not part of complementary or alternative therapies (this category is already in the review) but are part of the repertoire of traditional birthing systems.[4] Ignoring the existence of traditional practices perpetuates a biased view of self-care practices as interventions to

promote task-shifting and sharing from clinical settings to homes but neglects the potential of learning from other ways of knowing.

1 Raman S, Nicholls R, Ritchie J, et al. Eating soup with nails of pig: thematic synthesis of the qualitative literature on cultural practices and beliefs influencing perinatal nutrition in low and middle income countries. *BMC Pregnancy Childbirth* 2016;16:192. doi:10.1186/s12884-016-0991-z

2 Sarmiento I, Paredes-Solís S, Dion A, et al. Maternal health and Indigenous traditional midwives in southern Mexico: contextualisation of a scoping review. *BMJ Open* 2021;11:e054542. doi:10.1136/bmjopen-2021-054542

3 Raman S, Nicholls R, Ritchie J, et al. How natural is the supernatural? Synthesis of the qualitative literature from low and middle income countries on cultural practices and traditional beliefs influencing the perinatal period. *Midwifery* 2016;39:87–97. doi:10.1016/j.midw.2016.05.005

4 Jordan B. *Birth in four cultures: A crosscultural investigation of childbirth in Yucatan, Holland, Sweden, and the United States*. 4th ed. Long Grove, IL: : Waveland Press 1993.

Thank you for this comment. The review protocol specified that a broad range of self-care practices were eligible. Quantitative interventional studies/reviews of indigenous self-care practices, including the use of herbal products, traditional medicine, cultural psycho-social strategies, were potentially eligible. We note these four studies are qualitative studies/reviews and were thus not included in this review.

When considering the reviewer's comment, we hypothesise that indigenous self-care practices might be underrepresented due to a lack of effectiveness studies on these topics. We have added a discussion point to highlight this, as follows: "Last but not least, we found no eligible studies on traditional and indigenous self-care practices. While important qualitative research has been conducted on this topic [38,39], we hypothesise that there is a relative lack of studies exploring the health effects of these types of self-care practices. Future research on these topics could also provide new opportunities for improving cultural safety of maternity care services." (L306-310)

Abstract

R2Q3. Why are 45% of studies the majority?

We revised to "Many included studies..." (L36).

R2Q4. Indicate the included studies at the beginning of the results section.

Completed (L36).

R2Q5. The first two strengths are characteristics of a scoping review. They are not the strengths of this study.

We removed the first two strengths accordingly.

R2Q6. The abstract says the authors followed an inductive approach to categorisation, but in this section, they say they used categories from the WHO classification. Please correct this discrepancy. The inductive approach was used to categorise the interventions into 11 broad categories. The WHO Classification Framework was used to classify the venue of accessing self-care i.e. the location where patients were orientated to self-care or carried out self-care. To prevent confusion to readers, we have removed reference to WHO Classification Framework from the abstract and elaborated on how the WHO framework was used in the Methods (L147-149).

R2Q7. Exclusion criteria are not necessarily limitations of the study (P5, line 81). Please think of characteristics of the studies that will limit their conclusions within the scope of the review.

In responding to an earlier peer reviewer's comment, we have now acknowledged the overlap between primary studies and systematic reviews included in this scoping review (see R2Q1). We also acknowledge the lack of grey literature and non-English studies in our sample under Limitations (L341-344).

Methods

R2Q8. Please indicate the registry's date and the online platform's name and provide the link in brackets.

We added a statement confirming the overall last date of search (L97). The date of search for individual databases was provided in Supplementary File S3. The online platform's names were already provided (e.g. Ovid, Cochrane Library, EBSCOHost). We did not provide hyperlinks as the links to these interfaces are accessible via Google Search, and hyperlinks may not be compatible with the journal's electronic format.

R2Q9. Page 5, line 99. Please, consider reorganising the paragraph to connect the inclusion criteria to the first sentence.

We initially organised this section such that the first paragraph discusses the overall process, and the second paragraph details the inclusion/exclusion criteria, which were extensive. We understand that normally the last two sentences would be placed after the criteria. However, in this case, moving them to after the criteria will disrupt the flow of the first paragraph and create two single-sentence paragraphs.

R2Q10. Page 6, line 103. Please briefly clarify this criterion "Studies involving partners or family members in maternal care were also eligible."

We revised it to "Studies in which partners or family members carried out self-care for the pregnant woman, were also eligible" (L108).

R2Q11. Page 7, lines 129 to 131. Please revise for clarity.

We revised to the following: "Two reviewers independently extracted data using a standardised extraction form we developed to suit the review (Supplementary File S4). Data extracted included characteristics of each study's population, interventions, and outcomes measured". (L134-136)

R2Q12. Page 7, lines 137 to 142. Please, briefly elaborate to explain how you conducted the inductive thematic analysis. It is unclear why you used another set of categories for interventional and observational studies.

We added the following description of the inductive thematic analysis process, along with a reference to original article describing the method (Braun and Clarke, 2012). "An inductive thematic analysis process was undertaken [20]. After all self-care interventions were identified, we analysed their descriptions and identified themes among groups of interventions. For example, interventions involving taking dietary supplements or adjusting diet intakes were grouped under the theme "Diet & nutrition". These themes were refined until all interventions were accounted for, creating the final set of self-care categories." (L143-147).

We did not use another set of categories to classify the interventions. The WHO framework was used to classify venues of access to self-care, i.e. the setting or location where participants were orientated to or carried out self-care (see R2Q6). We elaborated this point to improve clarity (L148-149). We also explained why we applied this classification to interventional and observational studies only: "These information were recorded for primary studies only, as evidence synthesis studies may include studies with different characteristics." (L150-151).

R2Q13. Page 7, line 142. Please consider moving this sentence to the second place in the paragraph. We removed this sentence as it could create confusion to readers (see R2Q6).

R2Q14. Page 7, line 144. Please, briefly explain why you excluded complex interventions from the final analysis. Is this a limitation to mention after the abstract and in the discussion?

To clarify, we did not exclude complex interventions, rather we conducted and reported descriptive analysis for them separately. We acknowledge our initial wording was confusing in this regard, which we have revised. We have also added the following within the Limitations section to address this: "We reported the results for complex interventions (i.e. interventions with more than three different self-care practices) separately (Supplementary File S9), since segregating each complex intervention into multiple individual interventions would overestimate their representation in the sample and disregard potential synergy between the self-care components. Besides, because these studies did not introduce any new self-care intervention, or a substantive number of studies for any single self-care intervention that we had already identified, it is unlikely that including them into the main analysis would change our results and conclusion. Although the number of such studies is small (n=35, 6%), they represent how self-care can be implemented in real-life, and future reviews should study their effects using appropriate methods e.g. component network meta-analysis" (L319-328)."

R2Q15. Page 7, lines 145 to 147. However, you still need to explain what the final analysis was. What analyses did you do in R? Please, clarify

We revised the sentence to “R v4.1.2 was used to calculate the frequencies of various intervention characteristics, participant characteristics and outcomes for each self-care intervention, aggregate these frequencies by self-care categories and present them as bar charts.” (L155-158)

R2Q16. Please, explain how you handled the literature reviews to avoid duplication of results with other included studies or missing aspects of the studies that those reviews did not consider. This might be a significant limitation because you used the proportion of studies to report your results. Completed. Please refer to R2Q1 for the full answer.

Methods

R2Q17. Figure 2 was almost impossible to read. You may need to reorganise it. One option is to avoid duplicating the label of categories in module B). Also, you must indicate the number of studies for each row.

Thank you for pointing this out. We have reorganised the panels vertically and increased font size, as well as ensured all TIFF files are at least 300 PPI (>760 DPI) and 90x90mm in size. Please note that the figure's size and resolution tend to be scaled down when the system generates a PDF proof for reviewers. In case that it happens again, please refer to the TIFF files which show the figures in full resolution. We will discuss with the copyeditor and graphic designer for the best format for publishing, during the final proofing stage. In addition, we added a table with the number of studies and percentages in the Supplementary File.

We also did the same for Figure 4.

R2Q18. Page 8, line 166. Could you explain why the numbers do not add to 580? Is it due to literature reviews or studies not reporting a country? Did you consider the countries of the studies in the literature reviews? Please, be explicit about the denominator.

There are 163 studies that were not assigned to an income group, because they are evidence synthesis studies (systematic reviews and meta-analyses), and their included studies were not restricted to a geographic location. We added this line to clarify: “163 studies (28%) were reviews with no geographic restriction, and usually included studies from multiple countries” (L181-182).

R2Q19. Page 10, line 179. Why not include these studies in a category for complex interventions instead of eliminating them? Please, in the limitations, discuss the impact of this decision.

Completed. Please refer to R2Q14 for the full answer.

R2Q20. Page 9, lines 200 to 202. Is this potentially a result of the search strategy?

We believe that this is not caused by the search strategy, which was formulated to capture various self-care concepts and use very broad terms for maternal health care (see Supplementary File S3). Even if a study evaluates a campaign to promote awareness about self-care, without a specific health issue, it would be picked up by the search strategy for using terms describing self-care and maternal health. Instead, there are other possible reasons why there are few studies in this area: a population-level intervention is more difficult to evaluate in an RCT; outcomes related to programme coverage, behaviours and beliefs are more common than health outcomes; and interventions not targeting a specific health issue are more difficult to evaluate and be published.

R2Q21. Page 10, lines 207 to 210. There are no reasons to make these assumptions. Please only report that in 80 studies (15%), there was insufficient detail to determine that the aforementioned modes of access were used.

Completed.

R2Q22. Figure 4 is almost impossible to read. The authors should consider making this figure a table offered as an additional supplementary file.

Please refer to the full response in R2Q17.

R2Q23. Page 10, line 220. Please, provide the numbers for partner support in relation to peers.

Completed (L234-235).

R2Q24. Page 11, lines 232 to 233. Please, clarify that you used all the studies for this section. The note under Table 1 will not be enough to facilitate readers to see what you did. Also, please refrain from repeating the numbers and percentages that you already have in the table.

Completed (L245, “Among 580 included studies...”).

Since there is a large number of variables being measured and reported, we want to use the Results section as a place to organise and summarise the information from all tables and figures. We only reported the most important results; for example, for outcomes, we only reported the most commonly reported health outcomes, as well as mortality-related outcomes (clinically important) and behavioural outcomes (many self-care interventions target behavioural change).

Discussion

R2Q25. Page 13, lines 23 to 26. Do the authors consider health system and formal health care the same? Please revise the sentence.

We revised it to “As such, self-care should be viewed as an extension of, and not a separate entity to, formal healthcare services”. (L264-265).

R2Q26. Page 13, line 56. What elements do you have to think this likely reflects a more well-established evidence base for lifestyle modification interventions, generally associated with a lower risk of harm? The sentence might be unnecessary. Please, provide a reference or eliminate it. This sentence is our interpretation of the results. To elaborate, what we meant is that the larger number of studies on lifestyle interventions might be attributable to the fact that lifestyle interventions are perceived to be safer, thus trials on this type of interventions are preferable and relatively less complex to conduct. For greater clarity, we have revised the sentence to “This might be attributable to a perceived lower risk of harm when conducting trials investigating lifestyle interventions, compared to clinical or pharmacological interventions. Moreover, lifestyle interventions tend to take place in their own living environments and in their own time, and hence may be more amenable to self-care approaches.” (L280-283).

Strength and limitations

R2Q27. This section has similar problems to those described after the abstract. Please, take those comments into account in this section.

Completed. Please refer to R2Q1 for the full answer.

Reviewer: 3

Prof. Eric Steegers, Erasmus MC

The review was drafted by dr. Lenie van Rossem (l.vanrossem@erasmusmc.nl) and accorded by Prof. Eric A.P. Steegers

This manuscript concerns a scoping review on self-care interventions around (pre and post) pregnancy care. It is a relevant topic given the high burden on health care, where self-care can contribute in the solution for this. In general the paper is well-written, and the reporting –especially regarding the choices made and the methodology- is very thorough. Table 1, and figures 2 and 3 are well-designed to give a clear overview of the findings. My main feedback to improve the paper is about the clarity of the wording, and the structure/depth of the discussion. I first read the abstract, and then it was not completely clear to me what aspects were being reviewed. This became clear when I read the paper. Please see specific comments below.

Thank you for your feedback and comments to improve this paper.

Abstract

R3Q1. Objective&Results: I have some difficulty with the wording of the objective and results. It seems that the objective is to merely identify the number of apps for selfcare, categorized per period. By using the word evidence in the results, it might give the idea that the effectiveness of the app is being studied.

We have revised the word “evidence” to “studies” in the abstract. We also limited the use of the word “evidence” throughout the main article, replacing it with “studies” or “research”.

R3Q2. Results: Could there be a discrepancy between the number of included studies in the abstract (545) versus the number of included studies in the main paper (580)?

Out of the 580 included studies, 545 were included in the main frequency analyses. The other 35 studies featured a complex intervention; we added a line to note that these studies were analysed and reported separately in Supplementary File S9 (L319-321). Please also see our previous response (R2Q14), where we have revised the text to make it clearer how these 35 studies were handled.

Background

R3Q3. Some systematic reviews have been....pregnancy [9-11]': I would suggest to reword the sentence. It now seems that you are repeating those reviews, but they have actually been performed for specific outcomes. I think it is important to add that as a detail.

Not only did these studies focus on specific outcomes, the scope of the interventions was also narrow (e.g. Dalrympe et al., 2018 only investigated lifestyle interventions). Moreover, only prenatal and/or postpartum periods were studied. We added the following sentence to further explain this limit in scope: "however these previous reviews focused on a few selected interventions and outcomes, and mainly the antenatal or postpartum periods" (L74-75).

We believe that the next sentence explains adequately the research gap that we are trying to address, and that our review was not merely repeating previous reviews: "No previous review has explored the broad and heterogeneous body of research across preconception, antenatal, intrapartum and postpartum interventions, in order to identify those that may be effective or applicable from a self-care perspective." (L75-78).

Methods

R3Q4. How did you deal with potential overlap between systematic reviews and original studies? Please refer to R2Q1 for the full answer. In short, we have determined the number of overlapping (n=45, 8%) and added a footnote to Figure 3 and Supplementary File S10 to highlight this overlap.

Results

R3Q5. Type of outcomes: Can you extend upon the types of outcomes? I was a bit surprised to see some of those outcomes being used for self-care (such as Apgar scores?).

The full list of outcome categories and their definitions can be found in Supplementary File S5. Apgar scores were evaluated in studies of self-care interventions for pain management during labour, asthma management during pregnancy, probiotics intake to reduce preterm birth, and other similar interventions.

R3Q6. To me, the 'study design' aspects of the included studies seem not to be of added value as this review does not review the effectiveness of the apps, or the quality of the studies. Can you explain why you have included this?

One of the purposes of this review is to guide further evidence synthesis activities and assist with WHO guideline development in this research area. We considered study designs because these provide a more meaningful and accurate reflection of the state of research for each self-care intervention. Instead of reporting just the volume of evidence, we wanted these findings to reflect what kind of data are available: was there only data from primary studies, or was there also synthesis data from systematic reviews and meta-analyses? Knowing the number and the type of evidence available is useful for WHO and other guideline developers to make decisions about future research and/or readiness for guideline development - i.e. which intervention to further investigate in an RCT, or which intervention has sufficient primary data to warrant a systematic review or meta-analysis. (See R3Q10 to see how this will be useful for future research prioritisation).

Discussion

R3Q7. Main findings: the main findings start with a summary of the results, but from 'Primary healthcare services are....' it seems more about interpretation of the findings. I would suggest to separate these.

We have organised this section to improve coherence (L260-265).

R3Q8. Can you extend on the meaning of 'digital technologies'?

We have revised it to "digital platforms (e.g. websites, mobile apps)" (L271).

R3Q9. 'Most studies concentrated on self-care interventions related to lifestyle modification.... This likely reflect a more well-established evidence for lifestyle modification interventions, generally

associated with a lower risk of harm'. In addition, I think that lifestyle interventions are eminently suitable for self-care, as lifestyle takes place all day and in the own environment.

Thank you for your input. We added this point as a discussion point: "Moreover, lifestyle interventions tend to take place in their own living environments and in their own time, and hence may be more amenable to self-care approaches (L281-282).

R3Q10. Can you extend on your ideas for future research?

We have added a section ("Future research", L346-360) to discuss ideas for future research, and provide examples of self-care interventions that should be prioritised. We also elaborated on what type of evidence is needed for future research (primary studies vs reviews), which further illustrates the importance of considering study designs in our analysis.

R3Q11. I think a comparison with the previous 3 reviews that were referenced in the introduction can be a valuable addition to the Discussion.

Completed (L263-264). One of the reviews was a conference abstract and did not provide further details for comparison.

VERSION 2 – REVIEW

REVIEWER	Sarmiento, Ivan McGill University, Family Medicine
REVIEW RETURNED	16-Feb-2023

GENERAL COMMENTS	Thank you very much for the invitation to read the updated manuscript. The authors have addressed almost all my comments. I want to insist on two minor revisions. In the strengths and limitations section, the two initial strengths remain unclear. Is the first strength related to the broad scope of the review? I agree that this is a strength of the study, but the current statement looks more like the study's objective. The second strength is debatable. As you will see in my following comment, it still needs to be clarified how the authors used the WHO categories. However, I do see the effort of the authors to articulate the results with the WHO framework as a strength of the study. In their response to my initial comments (R2Q6), the authors explained that 'the WHO Classification Framework was used to classify the venue of accessing self-care.' And this is also in the methods. However, in the results (L207-208), they say, 'the identified interventions correspond to a range of categories from WHO's classification of self-care interventions'. In my opinion, there is no conflict between using pre-existing categories and identifying new ones from data. However, it is necessary to clarify what the authors did. Finally, I appreciated the final note regarding traditional and indigenous self-care practices. This is a field that well deserves more attention. Beyond the discussion on cultural safety, in previous work in Mexico, traditional midwives highlighted that, in their views, not following self-care practices was the most critical risk factor for unhealthy maternity in the remote communities where they live (1). (1) Sarmiento, I., Paredes-Solís, S., Loutfi, D. et al. Fuzzy cognitive mapping and soft models of indigenous knowledge on maternal health in Guerrero, Mexico. BMC Med Res Methodol 20, 125 (2020). https://doi.org/10.1186/s12874-020-00998-w
--

REVIEWER	Steeegers, Eric Erasmus MC
REVIEW RETURNED	10-Mar-2023

GENERAL COMMENTS	The authors have answered the questions in a satisfactory way and changed the manuscript accordingly in the revised version.
--

VERSION 2 – AUTHOR RESPONSE

Reviewer 2

Thank you very much for the invitation to read the updated manuscript. The authors have addressed almost all my comments. I want to insist on two minor revisions. In the strengths and limitations section, the two initial strengths remain unclear.

R2Q1. Is the first strength related to the broad scope of the review? I agree that this is a strength of the study, but the current statement looks more like the study's objective.

Yes, it is. We wanted to highlight the comprehensive scope as the main strength of the review. We elaborated on this point to make it clearer, as follows: "While there are systematic reviews for some self-care practices, this is the first scoping review that maps all potential self-care practices across the spectrum of pregnancy care, from preconception, antenatal, intrapartum to postnatal care. This scoping review provides a comprehensive summary of the state of research on this topic, and lays the foundation for future systematic reviews and primary research." (L318-320)

R2Q2. The second strength is debatable. As you will see in my following comment, it still needs to be clarified how the authors used the WHO categories. However, I do see the effort of the authors to articulate the results with the WHO framework as a strength of the study. In their response to my initial comments (R2Q6), the authors explained that 'the WHO Classification Framework was used to classify the venue of accessing self-care.' And this is also in the methods. However, in the results (L207-208), they say, 'the identified interventions correspond to a range of categories from WHO's classification of self-care interventions'. In my opinion, there is no conflict between using pre-existing categories and identifying new ones from data. However, it is necessary to clarify what the authors did.

We revised the Methods section as follows: "In addition, we matched the interventions to the categories listed in WHO's classification of self-care interventions for self-carers and caregivers [21]. We also recorded the venues of access to self-care, i.e. the setting or location where participants were orientated to or carried out self-care, based on this classification framework [21]." Given the WHO guideline that details the classification we used is quite novel (published 2021) and has not been otherwise applied in a systematic review, we consider it useful to acknowledge this as an addition value of the current review.

R2Q3. Finally, I appreciated the final note regarding traditional and indigenous self-care practices. This is a field that well deserves more attention. Beyond the discussion on cultural safety, in previous work in Mexico, traditional midwives highlighted that, in their views, not following self-care practices was the most critical risk factor for unhealthy maternity in the remote communities where they live (1).
(1) Sarmiento, I., Paredes-Solís, S., Loutfi, D. et al. Fuzzy cognitive mapping and soft models of indigenous knowledge on maternal health in Guerrero, Mexico. *BMC Med Res Methodol* 20, 125 (2020). <https://doi.org/10.1186/s12874-020-00998-w>

Thank you for your insight. We will continue to consult and draw from these studies in future research about traditional and indigenous self-care.

VERSION 3 – REVIEW

REVIEWER	Sarmiento, Ivan McGill University, Family Medicine
REVIEW RETURNED	10-Apr-2023
GENERAL COMMENTS	Thank you for addressing all my comments.